# Comparative efficacy of anti-vascular endothelial growth factor on diabetic macular edema diagnosed with different patterns of optical coherence tomography: A network meta-analysis

Jiajia Yao[1,2], Wanli Huang[3], Lixia Gao[2], Yan Liu[1], Qi Zhang[1], Juncai He[2,3], Li Zhang[1] *

1 The Second Affiliated Hospital of Chongqing Medical University, Chongqing, China, 2 Southwest Hospital/Southwest Eye Hospital, Third Military Medical University (Amy Medical University), Chongqing, P.R. China, 3 No. 927 hospital, Joint Logistics Support Force of Chinese PLA, Puer, Yunnan, China

* 202333@hospital.cqmu.edu.cn

**Data Availability Statement:** All relevant data are within the manuscript and its Supporting information files.

## Abstract

Intravitreal anti-vascular endothelial growth factor (anti-VEGF) injections have emerged as the most common therapeutic approach for the management of diabetic macular edema (DME). Despite their proven superiority over other interventions, there is a paucity of data regarding the relative effectiveness of anti-VEGF agents in treating DME diagnosed with different patterns of optical coherence tomography (OCT). In this regard, we conducted a systematic review and comparative analysis of the therapeutic efficacy of intravitreal bevacizumab, ranibizumab, aflibercept, and conbercept in the management of DME with diffuse retinal thickening (DRT), cystoid macular edema (CME), and serous retinal detachment (SRD) patterns identified using OCT. Our study encompassed a comprehensive search of PubMed, Embase, Web of Science, China National Knowledge Infrastructure (CNKI), and Wan Fang Data from their inception until January 25, 2023. The network meta-analysis involved the inclusion of 1606 patients from 20 retrospective studies with a moderate risk of bias but no evidence of publication bias. The DRT group had the highest increase in best-corrected visual acuity (BCVA) with anti-VEGF, while the SRD group had the greatest reduction in Central Macular Thickness (CMT). Furthermore, conbercept, ranibizumab, and bevacizumab, respectively, showed the best treatment outcomes for patients with DRT, CME, and SRD in terms of improvement in BCVA. And, conbercept exhibited the highest reduction in CMT in the DRT, CME, and SRD groups. In conclusion, our study highlights the efficacy of anti-VEGF agents in the management of DME and provides valuable insights into the selection of anti-VEGF agents tailored to the individual needs of patients.

**Funding:** The author(s) received no specific funding for this work.

**Competing interests:** The authors have declared that no competing interests exist.

## Introduction

Diabetic macular edema (DME) is a major cause of visual loss in patients with diabetes [1]. The pathogenesis of DME is thought to be attributable to alterations in the blood-retinal barrier, leading to the accumulation of fluid at the macula. Previous research has indicated heightened levels of inflammatory and angiogenic cytokines within the ocular environment [2]. Vascular endothelial growth factor (VEGF) induces angiogenesis and vascular hyperpermeability in DME by means of an inflammatory response[3] [3]. Therefore, treatment with anti-VEGF is currently one of the most promising approaches for treating vision loss due to DME [4].

The utilization of optical coherence tomography (OCT) technology has significantly contributed to the understanding of the morphological changes and intraretinal damage associated with DME. This diagnostic tool has been extensively employed in optimizing early intervention and monitoring the efficacy of therapies for macular edema [5]. As such, OCT has emerged as a vital tool in the management of DME, enabling timely and appropriate treatment to prevent further vision loss. During OCT examination, DME can present with several patterns including diffuse retinal thickening (DRT), cystoid macular edema (CME), and serous retinal detachment (SRD) [6]. Clinical studies have demonstrated that the efficacy of anti-VEGF treatment for DME may vary based on the optical coherence tomography (OCT) pattern observed, as each pattern is associated with a distinct pathogenesis [7, 8]. However, insufficient high-quality evidence exists to establish a definitive correlation between patterns observed with OCT and the effects of intravitreal anti-VEGF injections.

Over the last two decades, anti-VEGF drugs have become increasingly prevalent in the treatment of ocular diseases associated with retinal neovascularization and exudation, such as DME. The approved drugs include conbercept, bevacizumab, ranibizumab, and aflibercept, all of which are effective in addressing these conditions. The Federal Drug Administration (FDA) has approved the use of aflibercept and ranibizumab in the treatment of ocular conditions. While bevacizumab is only authorized by the FDA for the treatment of local and metastatic solid cancers, its off-label use in ocular conditions has been prevalent for over a decade [9]. Conbercept (KH902; Chengdu Kanghong Biotech Co., China) is a recombinant fusion protein with key domains 2, 3, and 4 from VEGF receptors 1 and 2, which is approved in China for the treatment of DME [10]. The potential variability in visual acuity benefits for DME among anti-VEGF drugs has been suggested by several individual trials and systematic reviews [11–14]. Currently, there is a limited number of meta-analyses that compare the clinical effects among anti-VEGF drugs (bevacizumab, ranibizumab, aflibercept, and conbercept) on patients with different patterns of OCT (DRT, CME, and SRD) in the management of DME.

To the best of our knowledge, only limited evidence has evaluated the comparison of the efficacy of treatment outcomes among different anti-VEGF drugs in DME. As such, it is crucial that healthcare professionals remain up-to-date with the latest research on DME treatments, to provide their patients with the best possible care. In light of the above information, it has been determined that a meta-analysis and systematic review of all available studies is necessary. The objective of this review is to assess the impact of OCT patterns on the treatment outcomes following intravitreal anti-VEGF therapy for DME. This comprehensive analysis will provide an updated and thorough understanding of the effectiveness of these therapeutic agents on the three morphologic patterns of DME as determined by OCT findings. Thus, in this study, we performed to derive evidence-based clinical guidelines for the anti-VEGF therapy in DME with different OCT patterns.

## Method

We performed a systematic review of publications on the use of anti-VEGF drugs for the treatment of DME. A synthesis of data inclusion followed the Preferred Reporting Items for Systematic Reviews and Meta-Analyses (PRISMA) guidelines.

### Search strategy

The search strategy was described in full in S1 Table. Two authors (Yao, and Huang) independently performed a systematic search. We searched with the terms "macular edema" OR "diabetes" AND the anti-VEGF agents (bevacizumab, ranibizumab, aflibercept, and conbercept) in the PubMed, Web of Science, Embase, Medline via Ovid, China National Knowledge Infrastructure (CNKI), and WanFang from the date of database inception to 25 January 2023, with no language restrictions. Reference lists of previous systematic reviews were also reviewed to identify additional eligible studies.

### Inclusion and exclusion criteria

Two authors (Yao, and Huang) independently reviewed all studies by title and abstract. After primary selection, two authors (Yao, and Huang) independently screened full-text studies, and considered for inclusion if they met the following criteria: (1) Including diabetes patients with DME; (2) With the intervention of anti-VEGF in preventing DME, including (bevacizumab, ranibizumab, aflibercept, and conbercept); (3) Reporting the change of Central Macular Thickness (CMT) related to baseline and/or the change of best-corrected visual acuity (BCVA) related to baseline. We excluded studies with the following criteria: (1) insufficient data for methodological quality assessment; (2) Reviews, editorials, letters, abstracts, case reports, or practice guidelines. Any disagreements about study inclusion/exclusion that could not be resolved by discussion between two authors (Yao, and Huang) were decided by a third author (Gao).

### Outcome measures

The following outcomes were quantitatively assessed: (1) the mean changes in BCVA from the baseline, indicating functional improvement; (2) the mean changes in CMT from the baseline, indicating anatomical improvement. BCVA recorded as ETDRS letters and Snellen fraction were transformed to Logarithm of the Minimum Angle of Resolution, (LogMAR) [15].

### Data extraction

Two authors (Yao, and Huang) independently extracted essential characteristics of included studies, including authors, country, the type of DEM, number of eyes, age, year of publication, outcomes, and intervention.

### Risk of bias assessment

All the included studies selected for meta-analysis were assessed independently by two observers according to the Newcastle-Ottawa Scale (NOS), a critical appraisal tool for retrospective studies [16]. The NOS consists of eight items within three sections: selection and definition of study groups (0–4 stars); comparability of study groups (0–2 stars); and outcome assessment and/or soundness of statistical analysis (0–3 stars). The total maximum score of these three subsets is 9. A study with a total NOS score of 7–9 was considered to be high quality, 4–6 indicated moderate quality and 1–3 indicated low quality.

## Statistical analysis

To compare the effects of each anti-VEGF agents for diabetic macular edema diagnosed with different patterns of optical coherence tomography, network meta-analysis (NMA) based on the Bayesian framework by integrating all available study results was conducted. All statistical analysis was performed using R Statistical Software (R Foundation for Statistical Computing, Vienna, Austria) and Stata software (version 15.1, StataCorp LLC, College Station, TX). 95% confidence intervals (CI) and a P value of < 0.05 were considered to be statistically significant. Statistical heterogeneity was assessed by the $I^2$ method with the chi-squared test ($I^2$ results between50 and 100% were considered to present significant heterogeneity). A fixed-effects model was applied to perform meta-analysis if $I^2 < 50\%$; otherwise, a random-effects model was used.

# Results

## Literature search

The process of identifying relevant studies was shown through the use of a flowchart, as depicted in Fig 1. Through systemic research, 8304 unique studies were identified. After reviewing the titles and abstracts of these articles, a further 8265 were excluded. Among the remaining 39 full-text studies, 20 of them into final network meta-analysis. The characteristics of the studies were summarized in Table 1. In all, a total of 1606 participants from 20 retrospective series were included in the final network meta-analysis. They were published between 2017 and 2020 and were mainly finished in China and Korea. All studies reported the patterns of DME recognized on OCT(DRT/CME/SRD) and used the intervention of intravitreal anti-VEGF agents injection. The follow-up duration, average number of injections, and medication dosage for all studies included in our meta-analysis were shown in S2 Table

## Quality assessment

The methodological quality of the individual studies, measured with the NOS, was shown in Fig 2. Overall, the quality score of the included studies ranged from 6 to 9 points. 17 studies were assessed as high quality (≥7 points) and 3 studies were assessed as moderate quality.

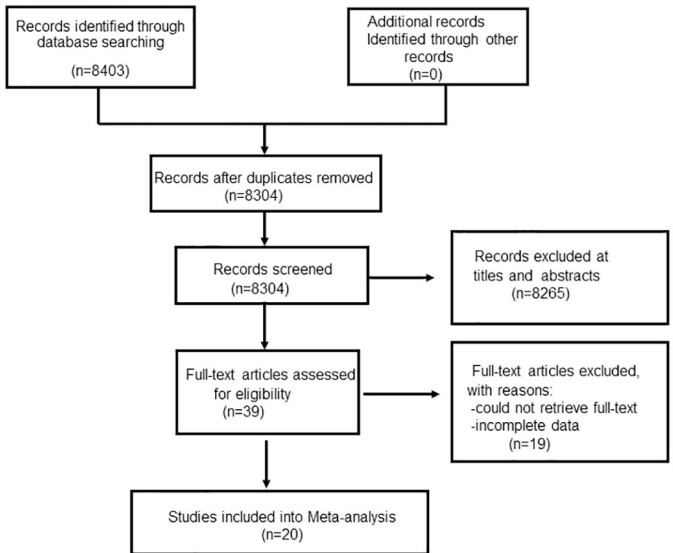

**Fig 1. Flowchart of the included study.**

**Table 1. Characters and results of included studies in meta-analysis.** Abbreviations: BCVA, best-corrected visual acuity; CME, cystoid macular edema; CMT, central macular thickness; DRT, diffuse retinal thickening; SRD, serous retinal detachment; VEGF, vascular endothelial growth factor.

| Author | country | year | The type of DEM | Number of Eyes | Age, year (Mean, SD) | Intervention | outcomes |
|---|---|---|---|---|---|---|---|
| Yijun Hu | China | 2019 | SRD | 113 | 58.2(13.8) | ranibizumab | BCVA/CMT |
| Xiao-Qing Li | China | 2017 | DRT/CME/SRD | DRT:34 CME:47 SRD:34 | DRT:60.68(18.18) CME:60.00(11.97) SRD:64.91(15.92) | conbercept | BCVA/CMT |
| Moosang Kim | Korea | 2011 | DRT/CME/SRD | DRT:29 CME:21 SRD:15 | DRT:60.25(10.42) CME:57.42(11.55) SRD:59.37(13.20) | bevacizumab | BCVA/CMT |
| Sehnaz Ozcaliskan | Turkey | 2020 | DRT/CME/SRD | DRT:37 CME:40 SRD:38 | DRT:64.10(7.73) CME:644.92(6.97) SRD:62.31(8.46) | aflibercept | BCVA/CMT |
| Mi In Roh | Korea | 2010 | DRT/CME/SRD | DRT:28 CME:28 | DRT:62.32(10.14) CME:64.25(5.87) | bevacizumab | BCVA/CMT |
| Haider R.Cheema | Ireland | 2014 | DRT/CME/SRD | DRT:20 CME:28 SRD:6 | DRT:53.2 CME:60.9 SRD:58.1 | bevacizumab | BCVA/CMT |
| Muhammad Atif Mian | Bahrain | 2015 | DRT | DRT:25 | DRT:58 | bevacizumab | BCVA/CMT |
| Nan-Ni Chen | China | 2020 | DRT/CME/SRD | DRT:36 CME:76 SRD:42 | DRT:64.60(9.20) CME:64.92(9.41) SRD:60.96(9.61) | ranibizumab | CMT |
| Pei-Chen Wu | China | 2012 | DRT/CME/SRD | DRT:10 CME:10 SRD:3 | DRT:64.60(7.01) CME:61.20(6.94) SRD:63.00(12.29) | bevacizumab | BCVA/CMT |
| Ahmed T.AL Sayed | Egypt | 2019 | DRT/CME/SRD | DRT:10 CME:10 SRD:10 | DRT:54.80(8.52) CME:54.60(15.41) SRD:54.00(8.87) | ranibizumab | BCVA/CMT |
| Sadhana Sharma | Nepal | 2022 | DRT/CME/SRD | DRT:40 CME:37 SRD:35 | DRT:55.88(7.39) CME:55.35(10.21) SRD:55.43(9.68) | bevacizumab | BCVA/CMT |
| Mouna Al Saad | Jordan | 2021 | DRT/CME/SRD | DRT:27 CME:24 SRD:5 | | Anti-VEGF | BCVA/CMT |
| KYUNG HOON SEO | Korea | 2016 | DRT/CME/SRD | DRT:23 CME:16 CME:16 | DRT:60.05(9.89) CME:54.91(11.60) SRD:56.92(14.29) | ranibizumab | BCVA/CMT |
| MASAHIKO SHIMURA | Japan | 2013 | DRT/CME/SRD | DRT:50 CME:38 SRD:25 | DRT:67.1(5.7) CME:66.5(5.5) SRD:64.6(5.1) | bevacizumab | BCVA/CMT |
| A Koytak | Tuekey | 2013 | DRT/CME/SRD | DRT:42 CME:31 SRD:19 | DRT:57.21(8.22) CME:59.29(11.73) SRD:58.95(10.99) | bevacizumab | BCVA/CMT |
| Yuan Ye | China | 2022 | DRT/CME/SRD | DRT:20 CME:20 SRD:20 | DRT:55.13(8.43) CME:56.32(8.12) SRD:(55.28(7.39) | ranibizumab | BCVA/CMT |
| Lu Yi | China | 2021 | DRT/CME/SRD | DRT:75 CME:53 SRD:31 | SRD:55.13(8.40) CME:56.64(9.34) SRD:55.24(8.34) | ranibizumab | BCVA/CMT |
| Bai Yang | China | 2021 | DRT/CME/SRD | DRT:28 CME:21 SRD:29 | SRD:58.3(8.67) CME:71.5(7.12) SRD:60.1(8.31) | conbercept | BCVA/CMT |
| Xue Yuanyuan | China | 2022 | DRT/CME/SRD | DRT:16 CME:27 SRD:14 | DRT:56.50(14.00) CME:57.00(27.00) SRD:58.50(4.00) | aflibercept | BCVA/CMT |
| Li Xiaoqing | China | 2018 | DRT/CME/SRD | DRT:20 CME:36 SRD:18 | DRT:60.68(18.18) CME:61.00(11.97) SRD:64.91(15.92) | conbercept | BCVA/CMT |

A

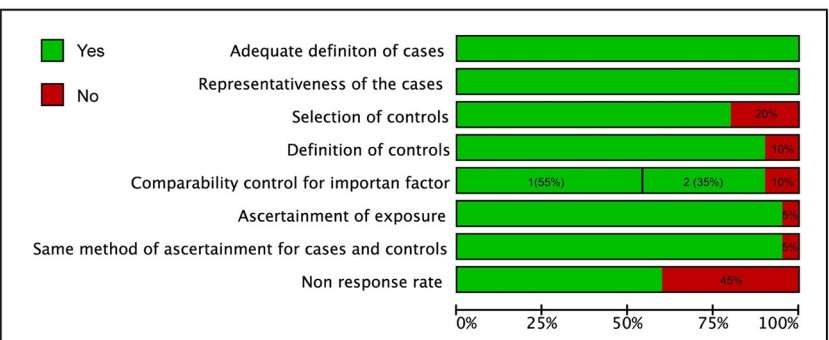

B

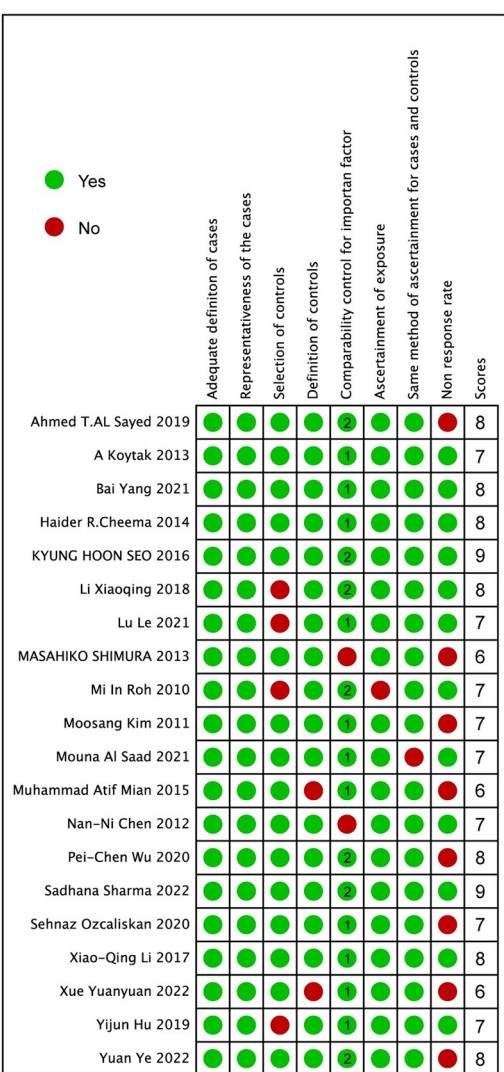

**Fig 2. Risk of bias graph (A) and summary (B) for each included study.**

## Effects of interventions

Because BCVA is the main visual index to judge the curative effect and progress, and CMT is an important anatomical index to judge the degree of macular edema, we analyzed the data of BCVA and CMT. Among these studies we have included, the baseline BCVA and CMT in S2 Table did not exactly match (S1 Fig). Therefore, we adopted the mean change in BCVA and CMT as the primary outcome. To address clinical heterogeneity, encompassing diverse follow-up timelines, average number of injections, and medication dosage administered, we conducted an extensive meta-regression analysis according to the Cochrane Handbook for Systematic Reviews of Interventions guidelines [17]. Our objective was to examine the impact of these variables on two critical outcomes: BCVA and CMT across various patient groups, which included DRT, CME, and SRD. Our findings, which were outlined in S3 Table, revealed that the variations in follow-up time, average number of injections, and medication dosage did not significantly influence the outcomes of BCVA and CMT across the six patient groups examined.

These studies indicated a noteworthy improvement in BCVA (S2A Fig) and a reduction of CMT (S2B Fig) in patients with DME when treated with anti-VEGF medications. The estimates for the treatment effect from the network meta-analysis (NMA) are presented in Table 2. The statistically significant increase in BCVA gained from baseline was found in anti-VEGF for DRT (MD = 0.16, 95%CI: 0.11 to 0.22), CME (MD = 0.15, 95%CI: 0.09 to 0.20), and SRD (MD = 0.12, 95%CI: 0.07 to 0.17) when compared to the sham group. The application of anti-VEGF therapy in the context of CMT was observed to be more efficacious than the sham group, which was demonstrated by a significant reduction in DRT (MD = 62.82, 95%CI: 39.97 to 89.08), CME (MD = 136.44, 95%CI: 109.29 to 163.49), and SRD (MD = 138.97, 95%CI: 111.22 to 166.80). The percentage probability of each type of DME treated by anti-VEGF being ranked first based on the change in BCVA and CMT were DRT (BCVA: 65.42%, CMT: 0.00%), CME (BCVA: 27.91%, CMT: 44.90%), and SRD (BCVA: 6.67%, CMT: 55.10%) respectively (Fig 3).

Furthermore, there was a notable variation in the effectiveness outcomes between anti-VEGF drugs for the patients with DRT (S3 Fig), CME (S4 Fig), and SRD (S5 Fig) respectively. Greater improvement in BCVA and a more significant reduction in CMT were observed in patients with DME treated with conbercept compared to those treated with ranibizumab, bevacizumab, and aflibercept (Table 3). Specifically, the mean changes in BCVA for the conbercept group were 0.04 (-0.16, 0.17), 0.09 (-0.06, 0.25), and 0.07 (-0.14,0.28) compared to

**Table 2. Results of network meta-analyses of efficacy outcomes (A and B) in diabetes patients with different OCT patterns relative to each other.** Abbreviations: BCVA, best-corrected visual acuity; CIs, confidence intervals; CMT, central macular thickness.

| Comparators | DRT | CME | SRD | Sham |
|---|---|---|---|---|
| **A) Chang in BCVA from baseline:** mean differences (95% CIs) | | | | |
| DRT | - | 0.02(-0.06,0.09) | 0.04(-0.03,0.12) | 0.16(0.11,0.22) |
| CME | -0.02(0.06,-0.09) | - | 0.03(-0.05,0.10) | 0.15(0.09,0.20) |
| SRD | -0.04(0.03,-0.12) | -0.03(0.05,0.10) | - | 0.12(0.07,0.17) |
| Sham | -0.16(-0.11,-0.22) | -0.15(-0.99,-0.20) | -0.12(-0.07,-0.17) | - |
| **B) Change in CMT from baseline in CMT:** mean differences (95% CIs) | | | | |
| DRT | - | -73.66(-110.97,-35.73) | -76.17(-114.09,-37.76) | 62.82(39.97,89.08) |
| CME | 73.66(110.97,35.73) | - | -2.53(-41.47,36.19) | 136.44(109.29,163.49) |
| SRD | 76.17(114.09,37.76) | 2.53(41.47,-36.19) | - | 138.97(111.22,166.80) |
| Sham | -62.82(-39.97,-89.08) | -136.44(-109.29,-163.49) | -138.97(-111.22,-166.80) | - |

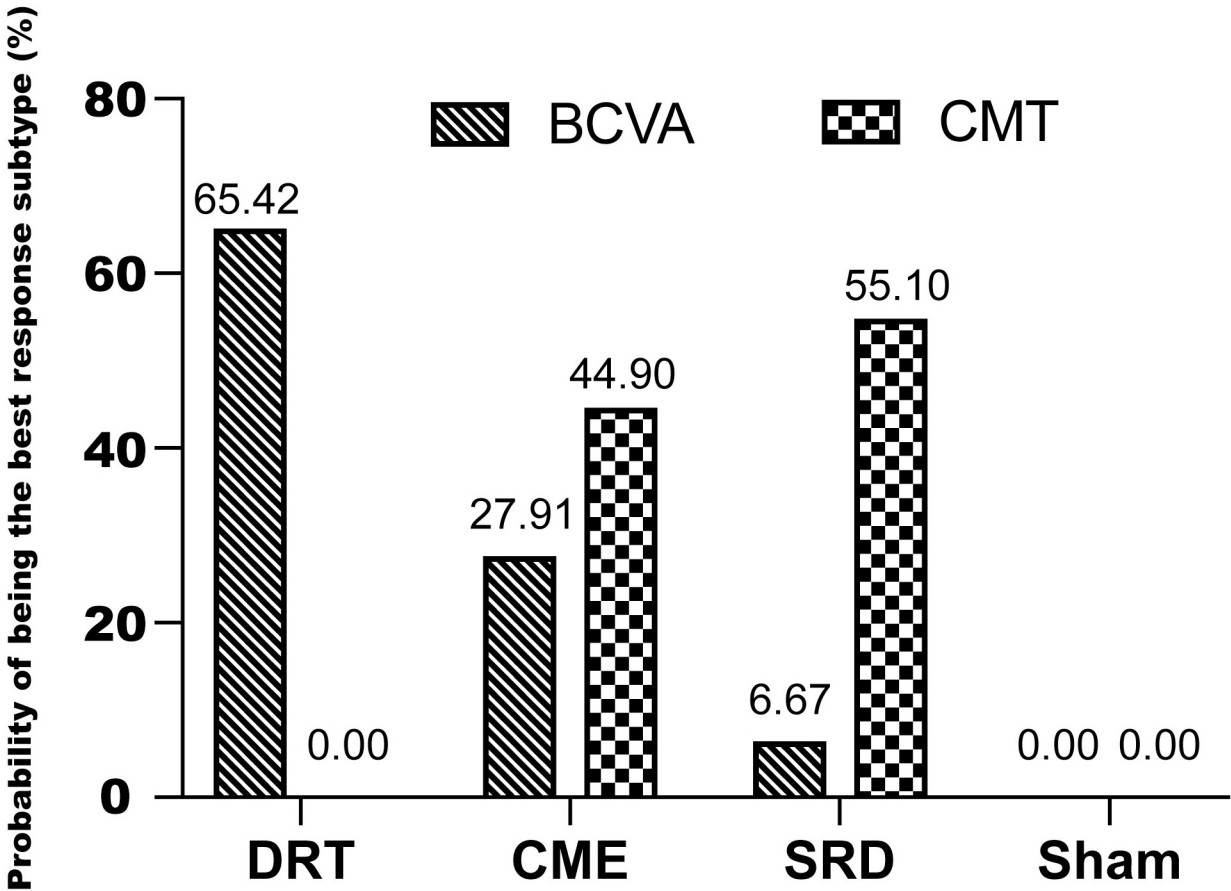

**Fig 3. Percentage probability of each type of DME being ranked first by outcome measure with the treatment of anti-VEGFs.** Abbreviations: BCVA, best-corrected visual acuity; CME, cystoid macular edema; CMT, central macular thickness; DRT, diffuse retinal thickening; SRD, serous retinal detachment; VEGF, vascular endothelial growth factor.

**Table 3. Results of network meta-analyses of efficacy outcomes (A and B) for all anti-VEGF drugs relative to each other under the DRT pattern.** Abbreviations: Afl, Aflibercept; BCVA, best-corrected visual acuity; Be, Bevacizumab; CIs, confidence intervals; CMT, central macular thickness; Con, Conbercept; DRT, diffuse retinal thickening; OCT, optical coherence tomography; Ran, Ranibizumab; Sha, sham.

| Comparators | Con | Ran | Be | Afl | Sha |
|---|---|---|---|---|---|
| **A) Chang in BCVA from baseline: mean differences (95% CIs)** | | | | | |
| **Con** | - | -0.04(-0.18,0.17) | -0.09(-0.25,0.06) | -0.07(-0.28,0.14) | -0.21(-0.34,-0.08) |
| **Ran** | 0.04(-0.16,0.17) | - | -0.85(-0.22,0.05) | -0.06(-0.26,0.14) | -0.21(-0.322,-0.10) |
| **Be** | 0.09(-0.06,0.25) | 0.08(-0.05,0.22) | - | 0.01(-0.17,0.21) | -0.12(-0.21,-0.03) |
| **Afl** | 0.07(-0.14,0.28) | 0.06(-0.14,0.26) | -0.01(-0.21,0.17) | - | -0.14(-0.32,0.02) |
| **Sha** | 0.21(0.08,0.34) | 0.21(0.10,0.32) | 0.12(0.03,0.21) | 0.14(-0.02,0.32) | - |
| **B) Change in CMT from baseline: mean differences (95% CIs)** | | | | | |
| **Con** | - | -38.934(-97.99,22.09) | -62.56(-117.69,-5.07) | -65.41(-135.58,5.54) | -106.58(-152.05,-62.16) |
| **Ran** | 38.93(-22.09,97.99) | - | -23.63(-75.96,29.18) | -26.39(-95.34,40.2) | -67.69(-19.42,-28.82) |
| **Be** | 62.56(5.07,117.69) | 23.63(-29.18,75.96) | - | -2.88(-67.75,60.05) | -44.06(-79.3,-12) |
| **Afl** | 65.41(-5.54,135.58) | 26.39(-40.2,95.34) | 2.88(-60.05,67.75) | - | -41.23(-95.92,12.82) |
| **Sha** | 106.58(62.16,152.05) | 67.69(28.82,109.42) | 44.06(12,79.3) | 41.23(-12.82,95.92) | - |

ranibizumab, bevacizumab, and aflibercept, respectively. The mean changes in CMT for the conbercept group were -38.934 (-97.99, 22.09), 62.56 (5.07, 117.69), and 65.41 (-5.54, 135.58) for the same respective drugs. Based on the change in BCVA (Fig 4A) and CMT (Fig 4B), the DTR group receiving conbercept exhibited the highest probabilities (BCVA: 44.42%, CMT: 88.43%) of being the most efficacious treatment.

The cohort of CME was analyzed (Table 4), and the results showed that conbercept (MD = 0.14, 95%CI: -0.01 to 0.29), ranibizumab (MD = 0.22, 95%CI: 0.08 to 0.357), bevacizumab (MD = 0.12, 95%CI: 0.01 to 0.22), and aflibercept (MD = 0.08, 95%CI: -0.11 to 0.28) displayed a statistically significant increase in BCVA when compared to the sham group. When it comes to CMT, it was found that anti-VEGF drugs resulted in a significant reduction compared to the sham groups. The MD for conbercept was -204.01 (95%CI: -259.14 to -149.92), for ranibizumab it was -114.76 (95%CI: -156.26 to -72.31), for bevacizumab it was -137.45 (95%CI: -175.3 to -95.77), and for aflibercept it was -94.34 (95%CI: -162.38 to -25.94). Furthermore, ranibizumab was found to be the most efficacious treatment with a probability of 69.49% for BCVA change (Fig 4C), while conbercept was the most efficacious treatment with a probability of 96.71% for CMT change (Fig 4D).

Table 5 showed the results of the DME with SRD pattern, in which conbercept (MD = 0.11, 95%CI: 0 to 0.22), ranibizumab (MD = 0.11, 95%CI: 0 to 0.21), bevacizumab (MD = 0.16, 95% CI: 0.05 to 0.27), and aflibercept (MD = 0.13, 95%CI: -0.04 to 0.31) exhibited a statistically significant increase in BCVA compared to the sham group. In terms of CMT, anti-VEGF drugs were significantly more effective than the sham group, with conbercept exhibiting the greatest reduction in CMT (MD = -215.86, 95%CI: -144.16 to -287.08), followed by bevacizumab (MD = -115.78, 95%CI: -55.28 to -176.89), ranibizumab (MD = -112.76, 95%CI: -62.62 to -164.21), and aflibercept (MD = -161.53, 95%CI: -66.37 to -259.11). The results of the study also indicate that bevacizumab had the highest probability of being the most efficacious treatment based on the change in BCVA (47.93%) (Fig 4E). Conbercept had the highest probability of being the most efficacious treatment based on the change in CMT (81.61%) (Fig 4F).

## Discussion

The treatment of DME with anti-VEGF therapy has proven to be effective in improving visual acuity and reducing macular edema [1]. In this study, we aimed to investigate whether different OCT configurations had an impact on the efficacy of anti-VEGF treatment for DME patients. The results of the study indicated that the group with DRT showed the highest improvement in BCVA with anti-VEGF treatment, while the group with SRD exhibited the most significant reduction in CMT. Additionally, the study evaluated the effectiveness of different anti-VEGF drugs in improving BCVA and reducing CMT. The results indicated that intravitreal conbercept was the most effective drug in reducing CMT and improving BCVA in patients with DRT. Furthermore, ranibizumab was found to be the most effective drug in improving BCVA in cases of CME, while conbercept was the most effective in reducing CMT. Finally, in the SRD group, bevacizumab proved to be the most effective in improving BCVA, while conbercept was the most effective in reducing CMT. Our data underscore the importance of considering the OCT characteristics when selecting an anti-VEGF agent for individual patients with DME.

The Diabetic Retinopathy Clinical Research Network (DRCRnet) has conducted research that confirms the preferred method of treatment for DME is through the use of anti-VEGF drugs [18]. This finding underscores the importance of utilizing the most effective treatments available to effectively manage this condition. Thus, the assessment of the efficacy of diverse DME treatments is of paramount importance for both ophthalmologists and policymakers.

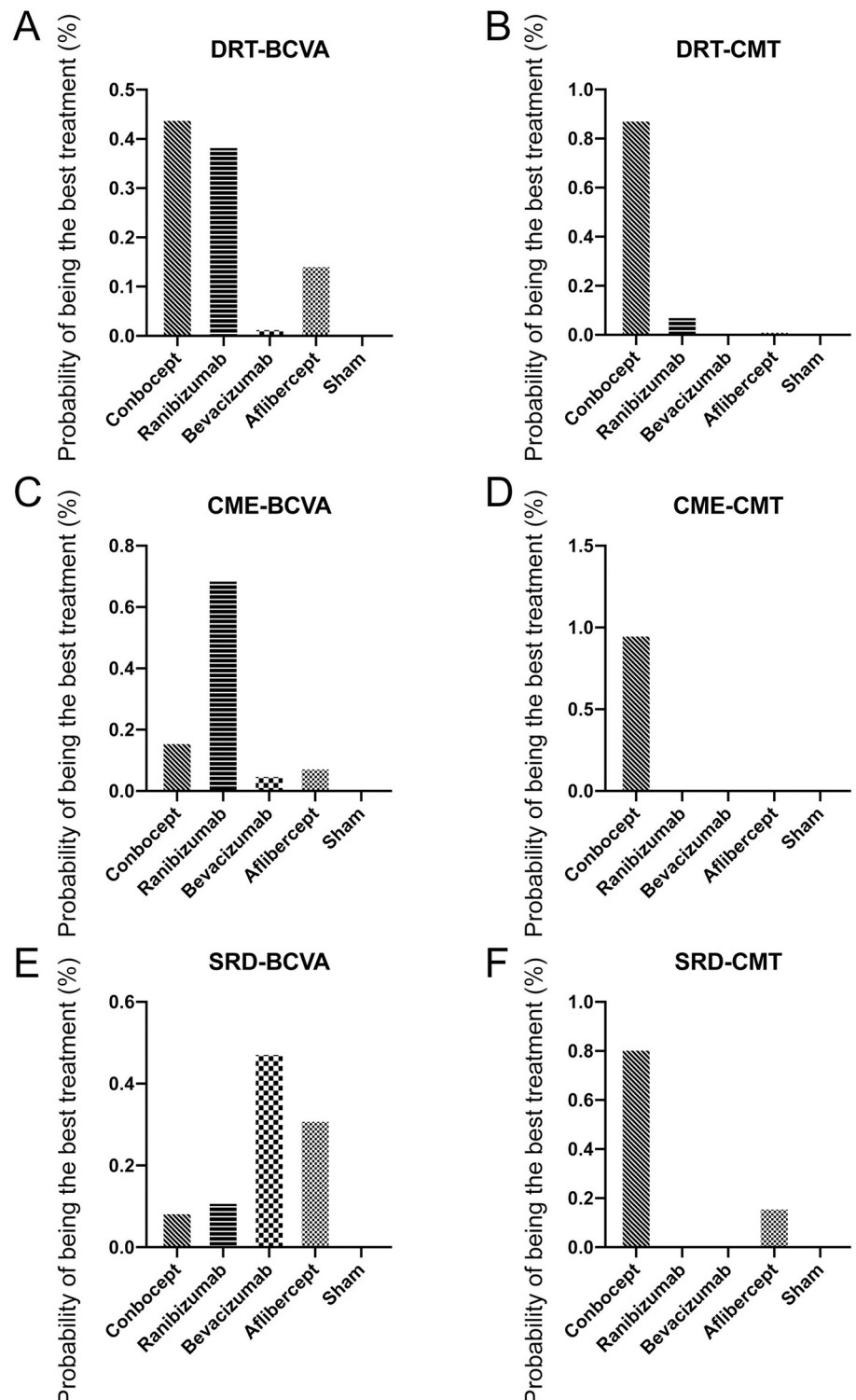

**Fig 4. Percentage probability of each treatment being ranked first by outcome measure for DME with DRT (A and B), CME (C and D) and SRD (E and F) patterns.** Abbreviations: BCVA, best-corrected visual acuity; CME, cystoid macular edema; CMT, central macular thickness; DRT, diffuse retinal thickening; SRD, serous retinal detachment; VEGF, vascular endothelial growth factor.

**Table 4. Results of network meta-analyses of efficacy outcomes (A and B) for all anti-VEGF drugs relative to each other under the CME pattern.** Abbreviations: Afl, Aflibercept; BCVA, best-corrected visual acuity; Be, Bevacizumab; CIs, confidence intervals; CME, cystoid macular edema; CMT, central macular thickness; Con, Conbercept; OCT, optical coherence tomography; Ran, Ranibizumab; Sha, sham.

| Comparators | Con | Ran | Be | Afl | Sha |
|---|---|---|---|---|---|
| **A) Chang in BCVA from baseline: mean differences (95% CI s)** | | | | | |
| **Con** | - | 0.07(-0.12,0.28) | -0.02(-0.20,0.16) | -0.05(-0.30,0.19) | -0.14(-0.29,0.01) |
| **Ran** | -0.07(-0.28,0.12) | - | -0.10(-0.27,0.07) | -0.13(-0.37,0.10) | -0.22(-0.35,-0.08) |
| **Be** | 0.22(-0.16,0.20) | 0.10(-0.07,0.27) | - | -0.03(-0.25,0.18) | -0.12(-0.22,-0.01) |
| **Afl** | 0.05(-0.19,0.30) | 0.13(-0.10,0.37) | 0.03(-0.18,0.25) | - | -0.08(-0.28,0.11) |
| **Sha** | 0.14(-0.01,0.29) | 0.22(0.08,0.357) | 0.12(0.01,0.22) | 0.08(-0.11,0.28) | - |
| **B) Change in CMT from baseline: mean differences (95% CI s)** | | | | | |
| **Con** | - | -89.27(-159.05,-21.34) | -66.48(-136.57,-1.27) | -109.51(-197.93,-23.19) | -204.01(-259.14,-149.92) |
| **Ran** | 89.27(21.34,159.05) | - | 22.75(-36.65,78.98) | -20.3(-100.63,59.9) | -114.76(-156.26,-72.31) |
| **Be** | 66.48(1.27,136.57) | -22.75(-78.98,36.65) | - | -43.08(-120.44,37.45) | -137.45(-175.3,-95.77) |
| **Afl** | 109.51(23.19,197.93) | 20.3(-59.9,100.63) | 43.08(-37.45,120.44) | - | -94.34(-162.38,-25.94) |
| **Sha** | 204.01(149.92,259.14) | 114.76(72.31,156.26) | 137.45(95.77,175.3) | 94.34(25.94,162.38) | - |

The efficacy of anti-VEGFs could be attributed to a range of factors, with the OCT biomarker emerging as a significant contributor [19]. OCT is a highly reliable imaging technique that allows for noninvasive and easy quantification of the CMT and accurate assessment of retinal anatomy [20]. Clinical data obtained from OCT has shown that anti-VEGF therapy can improve visual acuity, reduce CMT, and prevent vision decline in patients with DME by mitigating macular edema and exudation [21]. However, there are still controversies surrounding the anatomical and functional outcomes of anti-VEGF treatment for different OCT patterns of DME. Our analysis revealed that the mean change in BCVA was significantly better in the DTR group compared to the SRD and CME groups after anti-VEGF treatment. Nevertheless, the most significant reduction in CMT was observed in the SRD group.

Among many possible contributions regarding OCT, different key mechanisms involved in the development of specific types of DME based on OCT classification might explain how OCT configuration influenced the disease progression and therapeutic effects of anti-VEGF

**Table 5. Results of network meta-analyses of efficacy outcomes (A and B) for all anti-VEGF drugs relative to each other under the SRD pattern.** Abbreviations: Afl, Aflibercept; BCVA, best-corrected visual acuity; Be, Bevacizumab; CIs, confidence intervals; CMT, central macular thickness; Con, Conbercept; OCT, optical coherence tomography; Ran, Ranibizumab; Sha, sham; SRD, serous retinal detachment.

| Comparators | Con | Ran | Be | Afl | Sha |
|---|---|---|---|---|---|
| **A) Chang in BCVA from baseline: mean differences (95% CI s)** | | | | | |
| **Con** | - | 0.02(-0.15,0.20) | 0.07(-0.1,0.26) | 0.04(-0.17,0.28) | -0.08(-0.22,0.05) |
| **Ran** | -0.02(-0.2,0.15) | - | 0.05(-0.1,0.2) | 0.02(-0.18,0.23) | -0.11(-0.21,0) |
| **Be** | -0.07(-0.26,0.1) | -0.05(-0.2,0.1) | - | -0.03(-0.24,0.18) | -0.16(-0.27,-0.05) |
| **Afl** | -0.04(-0.28,0.17) | -0.02(-0.23,0.18) | 0.03(-0.18,0.24) | - | -0.13(-0.31,0.04) |
| **Sha** | 0.11(0,0.22) | 0.11(0,0.21) | 0.16(0.05,0.27) | 0.13(-0.04,0.31) | - |
| **B) Change in CMT from baseline: mean differences (95% CI s)** | | | | | |
| **Con** | - | -102.99(-189.74,-14.66) | -100.01(-193.1,-5.51) | -54.26(-173.03,67.02) | -215.86(-287.08,-144.16) |
| **Ran** | 102.99(14.66,189.74) | - | 2.935(-76.34,81.92) | 48.63(-58.86,158.02) | -112.76(-164.21,62.62) |
| **Be** | 100.01(6.51,193.1) | -2.93(-81.92,76.34) | - | 45.73(-67.16,160.57) | -115.78(176.89,-55.28) |
| **Afl** | 54.26(-67.02,173.03) | -48.63(-158.02,58.86) | -45.73(-160.57,67.16) | - | -161.53(-259.11,-66.37) |
| **Sha** | 215.86 (144.16,287.08) | 112.76(62.62,164.21) | 115.78(55.28,176.89) | 161.53(66.37,259.11) | - |

agents in DME [22]. OCT-based classification of DME can provide valuable insights into the structural and functional changes that occur in the retina and choroid and their correlation with clinical outcomes. The unique patterns observed in DME can have distinct pathogenic mechanisms. For instance, the occurrence of DRT is attributed to intracytoplasmic swelling of Müller cells arising from ischemia. Conversely, CME arises due to the necrosis of the Müller cells, resulting in the formation of cavities [23, 24]. SRD is a consequence of the dysfunction of the retinal pigment epithelium (RPE) and damage to the external limiting membrane (ELM) [25].

Several researchers have emphasized the sequential order that characterizes the development of DME, which involves the progressive damage of Müller cells leading to intracellular swelling, cyst formation, and liquid accumulation under the neurosensory retina. The breakdown of the inner blood-retinal barrier or damaged capillaries increases vascular permeability, leading to localized leakage and resulting in the DRT type change [26]. Several studies have shown that DRT has no cystoid degeneration or subretinal fluid. Anti-VEGF therapy suppresses vascular permeability, which seems to have a significant effect on treatment outcomes [27]. The DRT type has a better baseline BCVA and thinner CMT than the other types. Therefore, DRT is believed to be the earliest form of DME, and VEGF plays a major role in the development of edema, resulting in better treatment outcomes in cases of DRT. The deterioration of RPE function by inflammation or ischemic disorders may cause an accumulation of intraretinal fluid, leading to SRD [28, 29]. The disruption of ELM is most likely responsible for poor prognosis, as it may lead to accumulated fluid in the outer retina [30]. It suggests that CME and SRD share a common pathogenesis and that SRD usually precedes CME, following the reason that visual acuity in CME is significantly worse than that in SRD and foveal thickness is thicker in CME than in SRD [26]. This partially explains the better results of CMT reduction in the SRD group of our study, as well as the different outcomes between the groups. Finally, CME in diabetes has been associated with VEGF factors, prostaglandin, and inflammatory cytokines [30]. Therefore, anti-VEGF therapy alone may not be sufficient to eliminate CME or improve visual acuity. Understanding the key mechanisms involved in the development of specific types of DME based on OCT classification is essential for the effective management of the disease and better treatment outcomes.

Limited evidence comparing the efficacy of available anti-VEGF drugs also has an impact on decision-making on the prescription and reimbursement of drugs for patients with DME. Despite the potential benefits of anti-VEGF treatment, the associated costs are considerable, which underscores the importance of identifying the most effective drugs for different types of DME (as determined by OCT). Thus, this study secondly aimed to evaluate and compare the relative efficacy of anti-VEGF drugs in treating DME across various OCT patterns. The findings of this investigation indicated that conbercept, ranibizumab, and bevacizumab offer the most significant benefits based on BCVA improvement for DME with DRT, CME, and SRD, respectively. Additionally, conbercept was found to be the most effective in reducing CMT for DRT, CME, and SRD groups.

There is a debate as to whether there exists a significant disparity in the efficacy outcomes across all anti-VEGF drugs [11–13]. Some evidence suggests that in patients with poorer baseline visual acuity, aflibercept may confer an advantage over bevacizumab, albeit such evidence is limited to a subgroup analysis within a single DME trial and was not sustained over a two-year period [11]. One area that continues to be debated relates to whether the greater binding affinity of aflibercept and its potentially longer duration of action results in less frequent injections when compared to ranibizumab and bevacizumab, which is often cited as a rationale for its use, given the burden posed by frequent injections to both patient and clinician [13]. Bevacizumab and ranibizumab have been the most extensively studied agents. In general, the level

of evidence for aflibercept is lower, owing to it being a newer medication and only being evaluated in four of the included trials. Conbercept, similar to aflibercept, is a recombinant fusion protein composed of VEGF binding domain from human VEGF receptors 1 and 2 [31]. It has a high affinity for all VEGF isoforms and placental growth factors [32], which is mainly used in China. Of note, at the time of the study, aflibercept and ranibizumab had not been approved for the treatment of DME in China. Thus, additional clinical studies are needed to compare the efficacy of conbercept with other anti-VEGF agents in DME.

## Supporting information

**S1 Checklist. PRISMA 2020 checklist.**
(DOCX)

**S1 Fig. Baseline values for BCVA (A) and CMT (B).**
(DOCX)

**S2 Fig. Forest plot of BCVA (A) and CMT (B) outcomes in diabetes patients with different OCT patterns.**
(DOCX)

**S3 Fig. Forest plot of BCVA (A) and CMT (B) outcomes in diabetes patients with DRT patterns.**
(DOCX)

**S4 Fig. Forest plot of BCVA (A) and CMT (B) outcomes in diabetes patients with CME patterns.**
(DOCX)

**S5 Fig. Forest plot of BCVA (A) and CMT (B) outcomes in diabetes patients with SRD patterns.**
(DOCX)

**S1 Table. The search strategy.**
(DOCX)

**S2 Table. Detailed treatment information and baseline values of BCVA and CMT.**
(DOCX)

**S3 Table. Meta-regression analysis on BCVA and CMT outcomes.**
(DOCX)

## Author Contributions

**Conceptualization:** Li Zhang.

**Data curation:** Jiajia Yao, Wanli Huang, Lixia Gao, Qi Zhang, Juncai He, Li Zhang.

**Formal analysis:** Yan Liu.

**Investigation:** Jiajia Yao, Wanli Huang.

**Methodology:** Jiajia Yao, Wanli Huang, Qi Zhang.

**Project administration:** Jiajia Yao.

**Software:** Jiajia Yao, Lixia Gao, Yan Liu.

**Supervision:** Jiajia Yao.

**Validation:** Jiajia Yao.

**Writing – original draft:** Jiajia Yao, Yan Liu, Juncai He, Li Zhang.

**Writing – review & editing:** Li Zhang.

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
