## [Decision Letter · Decision Letter 0]

6 Feb 2024

PONE-D-23-36631Comparative efficacy of Anti-vascular endothelial growth factor on diabetic macular edema diagnosed with different patterns of optical coherence tomography: a network meta-analysisPLOS ONE

Dear Dr. Zhang,

Thank you for submitting your manuscript to PLOS ONE. After careful consideration, we feel that it has merit but does not fully meet PLOS ONE’s publication criteria as it currently stands. Therefore, we invite you to submit a revised version of the manuscript that addresses the points raised during the review process.

We look forward to receiving your revised manuscript.

Kind regards,

Tatsuya Inoue

Academic Editor

PLOS ONE

Reviewers' comments:

Reviewer's Responses to Questions

**Comments to the Author**

1. Is the manuscript technically sound, and do the data support the conclusions?

Reviewer #1: Yes

Reviewer #2: Partly

2. Has the statistical analysis been performed appropriately and rigorously? 

Reviewer #1: I Don't Know

Reviewer #2: Yes

3. Have the authors made all data underlying the findings in their manuscript fully available?

Reviewer #1: Yes

Reviewer #2: Yes

4. Is the manuscript presented in an intelligible fashion and written in standard English?

Reviewer #1: Yes

Reviewer #2: Yes

5. Review Comments to the Author

Reviewer #1: In this Meta-Analysis study, the authors rationally picked up some previous clinical studies investigating visual and morphological prognosis after anti-VEGF therapy according to morphological patterns, and then concluded DRT had the highest increase in BCVA, while SRD showed largest reduction of CMT. Generally, the significance of this study is high, and discussion about pathogenesis of each DME pattern is interesting.

One noted matter is that the authors overemphasized about the superiority of conbercept in abstract and discussion. In this review, comparative efficacies among each drug were not focused, thus description about each drug should be omitted, especially last paragraph in this manuscript.

Reviewer #2: In this study, the authors evaluated the comparative efficacy for the anti-VEGF therapy in DME with different OCT patterns. I consider this paper to be an interesting analysis of a vast amount of data, but with a number of major flaws.

1. Introduction: The authors cite a paper by Seo et al (2016). that classifies DME into three patterns. However, the 20 papers analyzed by the authors include meta-analysis of papers published before 2016, but are those cases correctly classified into the three categories? If the authors evaluate the treatment effect for each DME pattern according to the classification based on the Seo et al. paper, I believe that the meta-analysis should be limited to papers classified after 2016 according to the Seo et al. paper.

2. Results: As the authors stated, this study was not able to match the baseline visual acuity or CMT of each group. However, we need to know their baseline visual acuity and CMT values as reference values and should indicate them in the paper.

3. What are the anti-VEGF treatment regimens and the treatment and follow-up periods of the studies included in this meta-analysis? If these are different, the comparisons will be very meaningless.

6. PLOS authors have the option to publish the peer review history of their article (what does this mean?). If published, this will include your full peer review and any attached files.

Reviewer #1: No

Reviewer #2: No

---

## [Author Response · Author response to Decision Letter 0]

20 Mar 2024

Reviewer #1: In this Meta-Analysis study, the authors rationally picked up some previous clinical studies investigating visual and morphological prognosis after anti-VEGF therapy according to morphological patterns, and then concluded DRT had the highest increase in BCVA, while SRD showed largest reduction of CMT. Generally, the significance of this study is high, and discussion about pathogenesis of each DME pattern is interesting.

One noted matter is that the authors overemphasized about the superiority of conbercept in abstract and discussion. In this review, comparative efficacies among each drug were not focused, thus description about each drug should be omitted, especially last paragraph in this manuscript.

Reply 1: 

Thank you very much for your comments and suggestions.

In this review, we have only compared the variation in the effectiveness outcomes between anti-VEGF drugs for the patients with DRT (Supplemental Figure 3), CME (Supplemental Figure 4), and SRD (Supplemental Figure 5) respectively. As you mentioned, our results found that conbercept exhibited the highest reduction in CMT in the DRT, CME, and SRD groups, which was not able to provide enough evidence to support the superiority of conbercept in the treatment of DME. The comparative efficacies of each drug need more analysis in the future research. Thus, we have changed some sentences in this manuscript. Your comments made our conclusions more precise. 

Reviewer #2: In this study, the authors evaluated the comparative efficacy for the anti-VEGF therapy in DME with different OCT patterns. I consider this paper to be an interesting analysis of a vast amount of data, but with a number of major flaws.

1. Introduction: The authors cite a paper by Seo et al (2016). that classifies DME into three patterns. However, the 20 papers analyzed by the authors include meta-analysis of papers published before 2016, but are those cases correctly classified into the three categories? If the authors evaluate the treatment effect for each DME pattern according to the classification based on the Seo et al. paper, I believe that the meta-analysis should be limited to papers classified after 2016 according to the Seo et al. paper. 

Reply: 

Thank you very much for your comments and suggestions.

The paper by Seo et al (2016). that classifies DME into three patterns according to the paper by Otani, T. et al (1999). The 20 papers analyzed in this meta-analysis were published after 1999. Thus, we have replaced the paper by Otani, T. et al (1999) as the reference in our revised manuscript according to your suggestion.

2. Results: As the authors stated, this study was not able to match the baseline visual acuity or CMT of each group. However, we need to know their baseline visual acuity and CMT values as reference values and should indicate them in the paper.

Reply: 

Thank you very much for your comments and suggestions.

We have added the baseline value for BCVA or CMT of each group in Supplemental Figure 1 and in Supplemental Table 2 in our revised manuscript according to your suggestion. Your suggestion made our manuscript more professional.

3. What are the anti-VEGF treatment regimens and the treatment and follow-up periods of the studies included in this meta-analysis? If these are different, the comparisons will be very meaningless.

Reply: 

Thank you for your insightful question regarding the variability in anti-VEGF treatment regimens and the treatment and follow-up periods of the studies included in our meta-analysis. We recognize your concern that the inconsistency in these parameters could potentially affect the comparability of study outcomes. In response to this, we have meticulously detailed the follow-up duration, average number of injections, and medication dosage for all studies included in our meta-analysis in Supplemental Table 2. This comprehensive listing serves to present the basis for our analysis transparently.

Following the guidelines of the Cochrane Handbook for Systematic Reviews of Interventions1, we conducted an extensive meta-regression analysis to examine the impact of these variables on two critical outcomes: Best Corrected Visual Acuity (BCVA) and Central Macular Thickness (CMT) across different patient groups (DRT, CME, and SRD). Our findings, outlined in Supplemental Table 3, reveal that the variations in follow-up time, average number of injections, and medication dosage do not significantly influence the outcomes of BCVA and CMT across the six patient groups examined.

This detailed analysis underscores that, despite the clinical and methodological diversity among the included studies, the differences in treatment regimens and follow-up periods do not significantly alter the effects on BCVA and CMT outcomes. Such results are crucial in affirming the validity of our meta-analysis, suggesting that the differences in treatment specifics and follow-up durations observed across studies do not detract from our ability to derive meaningful conclusions from the aggregated data.

Therefore, while acknowledging the inherent heterogeneity in the studies we reviewed, our meta-regression analysis provides empirical evidence that this variability does not significantly impact the key outcomes of interest. This reinforces our belief that the results of our meta-analysis offer valuable insights into the effects of anti-VEGF treatments, despite the variability in treatment details and follow-up periods among the included studies.

We hope this comprehensive explanation, supported by the detailed enumeration in Supplemental Table 2 and backed by statistical analysis, adequately addresses your concerns. We are grateful for your thorough review and appreciate the opportunity to clarify this aspect of our research further. Additionally, we have included the meta-regression analysis details in the results and supplementary materials for your reference.

Reference：1. Deeks JJ, Higgins JPT, Altman DG (editors). Chapter 10: Analysing data and undertaking meta-analyses. In: Higgins JPT, Thomas J, Chandler J, Cumpston M, Li T, Page MJ, Welch VA (editors). Cochrane Handbook for Systematic Reviews of Interventions version 6.4 (updated August 2023). Cochrane, 2023. Available from www.training.cochrane.org/handbook.

---

## [Decision Letter · Decision Letter 1]

9 May 2024

Comparative efficacy of Anti-vascular endothelial growth factor on diabetic macular edema diagnosed with different patterns of optical coherence tomography: a network meta-analysis

PONE-D-23-36631R1

Dear Dr. Zhang,

We’re pleased to inform you that your manuscript has been judged scientifically suitable for publication and will be formally accepted for publication once it meets all outstanding technical requirements.

Kind regards,

Tatsuya Inoue

Academic Editor

PLOS ONE

Additional Editor Comments (optional):

Reviewers' comments:

Reviewer's Responses to Questions

**Comments to the Author**

1. If the authors have adequately addressed your comments raised in a previous round of review and you feel that this manuscript is now acceptable for publication, you may indicate that here to bypass the “Comments to the Author” section, enter your conflict of interest statement in the “Confidential to Editor” section, and submit your "Accept" recommendation.

Reviewer #2: All comments have been addressed

2. Is the manuscript technically sound, and do the data support the conclusions?

Reviewer #2: Yes

3. Has the statistical analysis been performed appropriately and rigorously? 

Reviewer #2: Yes

4. Have the authors made all data underlying the findings in their manuscript fully available?

Reviewer #2: Yes

5. Is the manuscript presented in an intelligible fashion and written in standard English?

Reviewer #2: Yes

6. Review Comments to the Author

Reviewer #2: The authors have added details of the meta-regression analysis in the results and supplementary materials, and I believe they have adequately responded to my comments.

7. PLOS authors have the option to publish the peer review history of their article (what does this mean?). If published, this will include your full peer review and any attached files.

Reviewer #2: No

---

## [Editor Report · Acceptance letter]

15 May 2024

PONE-D-23-36631R1 

PLOS ONE

Dear Dr. Zhang, 

I'm pleased to inform you that your manuscript has been deemed suitable for publication in PLOS ONE. Congratulations! Your manuscript is now being handed over to our production team.

Kind regards, 

on behalf of

Dr. Tatsuya Inoue 

Academic Editor

PLOS ONE